# Hemicellulosic Bioethanol Production from Fast-Growing Paulownia Biomass

Elena Domínguez [1], Pablo G. del Río [2], Aloia Romaní [2,*], Gil Garrote [2] and Lucília Domingues [3]

1 Technological Centre of Multisectoral Research (CETIM), Business Park of Alvedro, 15181 Culleredo, Spain; edominguez@cetim.es
2 Department of Chemical Engineering, Faculty of Science, Universidade de Vigo, Campus Ourense, As Lagoas, 32004 Ourense, Spain; pdelrio@uvigo.es (P.G.d.R.); gil@uvigo.es (G.G.)
3 CEB-Centre of Biological Engineering, Campus of Gualtar, University of Minho, 4710 057 Braga, Portugal; luciliad@deb.uminho.pt
* Correspondence: aloia@uvigo.es

**Abstract:** In order to exploit a fast-growing Paulownia hardwood as an energy crop, a xylose-enriched hydrolysate was obtained in this work to increase the ethanol concentration using the hemicellulosic fraction, besides the already widely studied cellulosic fraction. For that, *Paulownia elongata x fortunei* was submitted to autohydrolysis treatment (210 °C or $S_0$ of 4.08) for the xylan solubilization, mainly as xylooligosaccharides. Afterwards, sequential stages of acid hydrolysis, concentration, and detoxification were evaluated to obtain fermentable sugars. Thus, detoxified and non-detoxified hydrolysates (diluted or not) were fermented for ethanol production using a natural xylose-consuming yeast, *Scheffersomyces stipitis* CECT 1922, and an industrial *Saccharomyces cerevisiae* MEC1133 strain, metabolic engineered strain with the xylose reductase/xylitol dehydrogenase pathway. Results from fermentation assays showed that the engineered *S. cerevisiae* strain produced up to 14.2 g/L of ethanol (corresponding to 0.33 g/g of ethanol yield) using the non-detoxified hydrolysate. Nevertheless, the yeast *S. stipitis* reached similar values of ethanol, but only in the detoxified hydrolysate. Hence, the fermentation data prove the suitability and robustness of the engineered strain to ferment non-detoxified liquor, and the appropriateness of detoxification of liquor for the use of less robust yeast. In addition, the success of hemicellulose-to-ethanol production obtained in this work shows the Paulownia biomass as a suitable renewable source for ethanol production following a suitable fractionation process within a biorefinery approach.

**Keywords:** hemicellulosic ethanol; fast-growing species; inhibitors; industrial yeast; xylose fermentation; *Scheffersomyces stipitis*; *Saccharomyces cerevisiae*

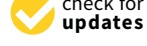

## 1. Introduction

The depletion of fossil resources and the increase of environmental concerns related to the $CO_2$ emissions are promoting the production of biofuels from lignocellulosic materials. However, the processing of this biomass to obtain biofuels, as ethanol, requires costly steps of operation. In this sense, fermentation of all sugars (including xylose from hemicellulosic fraction) would improve the economics of the process by 25% [1–3]. The xylose fermentation is considered one of the major challenges in lignocellulose-to-ethanol pathway [4] due to several factors. Firstly, hemicellulosic hydrolysates contain degradation compounds that may likely hinder the subsequent saccharification and fermentation process, such as furan derivatives (furfural and hydroxymethylfurfural), phenolic compounds and weak organic acids (acetic, levulinic and formic acids) [5–7]. In addition, the most used microorganism for industrial production of ethanol, *Saccharomyces cerevisiae* (Meyen ex E.C. Hansen 1883), is not naturally able to ferment pentoses [8–10]. Among the few yeast strains that can naturally ferment xylose, *Scheffersomyces stipitis* ((Pignal) Kurtzman and M. Suzuki 2010), is one of the most studied, being able to reach high ethanol yields [11]. Albeit, this yeast is

highly sensitive to inhibitors, which can hugely impede its growth, and ethanol production feasibility [12].

In order to overcome these difficulties, the removal of inhibitory compounds from the hemicellulosic hydrolysates by chemical, biological or physical methods is essential [6,13]. Detoxification strategies are the most employed and efficient procedures to tackle inhibition problems, namely evaporation, overliming, and liquid–solid extraction (activated charcoal, ion exchange) [13,14]. The main disadvantage of employing these procedures is the increase in the cost due to the additional separation steps to be made, and therefore, alternative strategies to surpass the inhibitory effect of degradation compounds have to be considered. In this context, the selection of microorganisms from natural or industrial environments with high resistance to inhibitors has emerged as an interesting solution [6,15]. Moreover, the inhibitor resistance of screened microorganism can be improved by evolutionary engineering [16] and/or genetic engineering [17,18] of previously identified molecular resistance determinants [19,20]. Additionally, the importance of the yeast chassis has been demonstrated as robust *S. cerevisiae* strains isolated from industrial environments metabolically engineered with the same xylose pathway have shown different outcomes for fermentation of non-detoxified hemicellulosic hydrolysates depending on the yeast background and hydrolysate [21].

With the purpose of obtaining a stream enriched in hemicelluloses (especially xylose), autohydrolysis processing followed by an acid-catalyzed step has been proposed as an efficient method for several feedstock, such as eucalyptus wood, corn cob, or vine trimming [22–24]. These hemicellulosic hydrolysates have been used for the biotechnological production of biofuels as ethanol [25] and value-added building blocks (such as xylitol and lactic acid) [23,24]. In addition, these processes are also applied for the production of furan compounds and furan derivatives (such as levulinic acid and formic acid), which are receiving more attention for their interesting features as components of fuels [26,27].

The raw material employed in this study was Paulownia wood, a fast-growing species with interesting features as renewable energy resource due to its high biomass production (50 t/ha·year) and tolerance to abiotic stress conditions [28,29]. Cellulosic fraction of Paulownia wood was already employed for bioethanol production via a sequential two-stage autohydrolysis [30] and through the combination of autohydrolysis and organosolv [31], or for pulping purposes via soda–anthraquinone processing [29]. However, in order to exploit Paulownia wood for energy purposes the employment of the totality of polysaccharides present in this feedstock is essential to improve the feasibility of the process. As far as the authors know, the acid-catalyzed depolymerization of the hemicellulosic fraction into value-added compounds has not been studied.

This study aims to evaluate, for the first time, the acid hydrolysis and fermentation of hemicellulose from Paulownia wood to obtain ethanol. For that, autohydrolysis pretreatment was proposed for xylan solubilization in the liquid phase and several steps of acid xylooligosaccharides hydrolysis, concentration, and detoxification were assessed to improve the ethanol production. Moreover, two yeast (natural xylose consuming and metabolic engineered strains) were used and compared for ethanol production from the hemicellulosic non-detoxified and detoxified Paulownia hydrolysates.

## 2. Materials and Methods

### 2.1. Raw Material

The raw material used in this work was *Paulownia elongata x fortunei* wood, kindly provided by a local wood producer (NW Spain). The feedstock was milled to a particle size of 8 mm and stored in a dry and dark place.

### 2.2. Preparation of Hemicellulosic Hydrolysate: Autohydrolysis and Dilute Acid Treatments

Figure 1 shows the process proposed and operational conditions used for the preparation of Paulownia hydrolysate. Briefly, hemicellulosic hydrolysate was obtained by autohydrolysis processing of Paulownia wood with water at liquid to solid ratio (LSR) of

6 g of water per g of biomass. Besides the temperature, the autohydrolysis processing can be expressed by means of its hardness using the severity ($S_0$), which can be calculated by the following equation:

$$S_0 = \log R_0 = \log(R_{0_{HEATING}} + R_{0_{COOLING}}) =$$
$$= \log\left(\left[\int_0^{t_{MAX}} \exp \cdot \left(\frac{T(t) - T_{REF}}{\omega}\right) \cdot dt\right] + \left[\int_{t_{MAX}}^{t_F} \exp \cdot \left(\frac{T\prime(t) - T_{REF}}{\omega}\right) \cdot dt\right]\right) \tag{1}$$

where $R_0$ corresponds with the severity factor, $t_{MAX}$ (min) corresponds with the time needed to reach the target temperature $T_{MAX}$ (°C), $t_F$ (min) corresponds with the time employed in the whole heating–cooling process, and $T(t)$ and $T\prime(t)$ correspond with the temperature profiles in the heating and cooling stages, respectively. The severity was quantified using the commonly reported values for $T_{REF}$ and $\omega$ (100 °C and 14.75 °C, respectively).

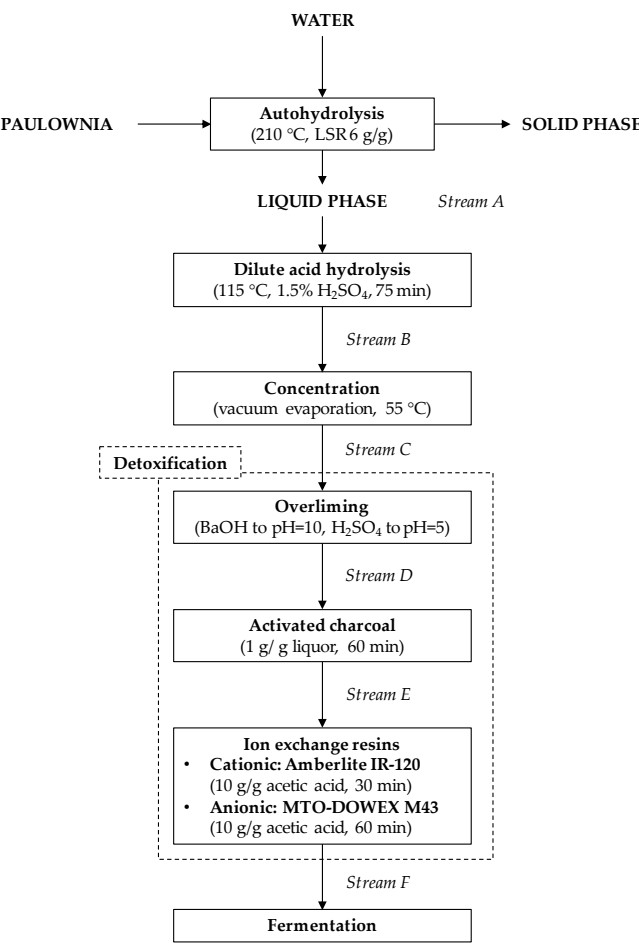

**Figure 1.** Flowchart of main processing steps to obtain a xylose-enriched stream derived from hemicellulosic fraction.

Autohydrolysis treatment was carried out in a pressurized stainless reactor (Parr Instruments Company, Moline, IL, USA) of 3.75 L internal volume at temperature of 210 °C ($S_0$ of 4.08), condition previously evaluated by Dominguez and collaborators [30], where maximum values for the recovery of xylooligosaccharides in the liquid phase were achieved. After the hydrothermal treatment, autohydrolysis liquor was separated from the solid phase by vacuum filtration. One aliquot of autohydrolysis liquor was analyzed directly to determine the concentration of monosaccharides, phenolic compounds, weak acids and furan compounds. Second aliquot of autohydrolysis liquor was submitted to quantitative acid post-hydrolysis (4% $H_2SO_4$, 121 °C for 40 min) for oligosaccharides quantification. For the preparation of the Paulownia hydrolysate, oligosaccharides present

in the autohydrolysis liquor were hydrolyzed by dilute acid hydrolysis in pressurized stainless reactor of 0.6 L internal volume with 0.5% of $H_2SO_4$ ($w/w$) at 125 °C for 180 min and 1.5% of $H_2SO_4$ ($w/w$) at 115 °C for 120 min [22]. With the aim of evaluating the acid dilute hydrolysis treatment, samples were withdrawn at desired times and analyzed for quantification of monosaccharides, furan, and weak acid concentration.

### 2.3. Concentration of Hemicellulosic Hydrolysate

Hemicellulosic hydrolysate (after dilute acid hydrolysis) was concentrated by a vacuum rotary evaporator. During the concentration stage, the boiling temperature of the hemicellulosic hydrolysate was maintained at 55 °C [8] to reach a final xylose concentration of 60 g/L. Concentrated hydrolysate was analyzed for sugars, weak acids and furans content by HPLC as same conditions that will be explained in Section 2.7.

### 2.4. Detoxification of Hemicellulosic Hydrolysate: Overliming, Activated Charcoal, and Ion Exchange

Concentrated hemicellulosic hydrolysate was detoxified to remove inhibitory compounds by the following stages. Firstly, concentrated hydrolysate was subjected to overliming process as described in Srilekha Yadav and colleagues [32] with slight modifications. Briefly, $Ba(OH)_2$ was added to hydrolysate to reach pH = 10 and incubated for 30 min followed by filtration and centrifugation to remove the precipitated matter. After that, pH was adjusted to 5 with $H_2SO_4$. Secondly, hydrolysate resulted from overliming process was treated with activated charcoal (at a 1:10 $w/w$ ratio of activated charcoal per hydrolysate) with agitation for 60 min at room temperature [33] to reduce the phenolic compounds content. Finally, the hemicellulosic hydrolysate was treated with ion exchange resins for acetic acid elimination [25]. Concentrated and neutralized hydrolysate was mixed with Amberlite IR-120 (cationic) resin in $H^+$ form at ratio 10 g of cationic resin per 1 g of acetic acid in the hydrolysate for 30 min and at room temperature in batch mode with agitation. Cationic resin was recovered by filtration and the hydrolysate was treated with MTO-DOWE X M43 (anionic) resin in $OH^-$ form at ratio 10 g of anionic resin per g of acetic acid present in the hydrolysate, in batch mode, with agitation for 60 min at room temperature. Hydrolysate treated with ion exchange resins was analyzed for sugars, furans, phenolic compounds, and weak acid content by HPLC as same conditions that will be explained in Section 2.7.

### 2.5. Yeast Species and Preparation of the Inoculum

Two yeast species were employed in this work: (i) *Scheffersomyces stipitis* CECT (Spanish Type Culture Collection) 1922, a natural xylose consumer, obtained from the Spanish Collection of Type Cultures (Valencia, Spain) and (ii) *Saccharomyces cerevisiae* MEC1133, an industrial strain previously engineered with the xylose metabolic pathway [34]. Cells were grown in a sterile solution containing 20 g xylose/L, 20 g peptone/L and 10 g yeast/L, which was previously sterilized a 112 °C for 20 min in an autoclave. The medium was placed in an orbital incubator at 30 °C and 200 rpm for 24 h. After that time, cells were recovered by centrifugation (15 min 4000× $g$, 4 °C) and resuspended in 0.9% NaCl to achieve a final concentration of 200$g$ of fresh yeast/L. The fermentation assays were inoculated with a final cell concentration of 1.5 g/L (quantified by dry cell weight).

### 2.6. Hemicellulosic Hydrolysate Fermentation

Fermentation assays were carried out in Erlenmeyer flasks placed in an orbital incubator at 30 °C and 100 rpm under oxygen-limited conditions (flasks closed with cotton stopper). Fermentation media were prepared from the concentrated hemicellulosic hydrolysate (either detoxified or non-detoxified), and diluted when necessary in order to reduce the inhibitors concentration (although decreasing the xylose concentration) to get a 100, 80, 60, or 50% of the concentration, referred to the concentrate hydrolysate (xylose concentration of 60 g/L). The media were sterilized by filtration (0.22 μm) and supplemented with nutrients (previously sterilized at 121 °C for 15 min), namely peptone and yeast extract at a final concentration of 20 and 10 g/L, respectively. The hydrolysates were

sampled at desired times and subjected to HPLC analysis to determine sugars consumption and ethanol production.

In addition, in order to compare fermentation profiles obtained by two yeast species, the volumetric productivity ($Q_P$, g ethanol/(L·h)) was calculated as follows:

$$Q_P = \frac{(Et)_t}{t} \tag{2}$$

where $(Et)_t$ is the ethanol concentration (g/L) at a time t (hours).

*2.7. Analytical Methods and Composition of the Raw Material*

Paulownia wood was analyzed for cellulose (measured as glucan), hemicellulose (xylan, arabinan and acetyl groups), Klason lignin, ethanol extractives, moisture and ash content following NREL (National Renewable Energy Laboratory) procedures [35–38]. Uronic acids were measured by and spectrophotometric method by Blumenkrantz and Asboe-Hansen [39]. The composition (expressed as g of component per 100 g of raw material in oven-dry basis ± standard deviations based on three replicate determinations) was: 39.7 ± 0.9 of glucan, 14.7 ± 0.35 of xylan, 3.9 ± 0.07 of acetyl groups, 23.9 ± 0.29 of Klason lignin, 7.35 ± 0.06 of extractives, 0.51 ± 0.03 of ashes, and 1.30 ± 0.30 of uronic acids (represented as glucuronic acid equivalent).

Hemicellulosic hydrolysate after different stages of processing (see Figure 1) and samples from hydrolysate fermentations were analyzed by HPLC for determination of oligosaccharides (glucooligosaccharides, xylooligosaccharides, arabino-oligosaccharides, and acetyl groups linked to oligosaccharides) via acid posthydrolysis 4% *w/w* $H_2SO_4$ at 121 °C for 40 min, monosaccharides (glucose, xylose, and arabinose), weak acids (acetic acid, levulinic acid, and formic acid), furan compounds (furfural and hydroxymethylfurfural), and ethanol concentration using an Aminex HPX-87H column (Bio-Rad, Hercules, CA, USA) at 50 °C and refractive index detector at 40 °C. The mobile phase (0.003 M $H_2SO_4$) was eluted at a flow rate of 0.6 mL/min. Total phenolic compounds (expressed as gallic acid equivalents) were also determined following the method described by Conde et al. [40].

*2.8. Analytical Methods and Composition of the Raw Material*

The evaluation of the acid posthydrolysis of the Paulownia hydrolysate was performed in duplicate, and results of the component concentration were presented graphically as mean values and standard deviation (SD) through error bars. On the other hand, fermentation assays were carried out in duplicate and the maximum ethanol concentration and maximum ethanol yield were presented as mean values ± SD. Statistical analysis was carried out using the software R (version 4.0.2). Differences among xylose concentrations at different times for each hydrolysis profile, and maximum ethanol concentrations and maximum ethanol yields in the fermentations for each yeast strain were tested with a one-way analysis of variance (ANOVA), followed by Tukey's test. Differences were considered as statistically significant when $p < 0.02$.

**3. Results and Discussion**

*3.1. Autohydrolysis Treatment of Paulownia*

As previously tested by the authors, autohydrolysis of *Paulownia elongata x fortunei* was a suitable processing to release a high amount of hemicellulosic derived compounds (especially xylooligosaccharides) in the liquid stream, while increasing the enzymatic susceptibility of the cellulose present in the residual solid phase [30]. Based on that work, Paulownia wood was subjected to autohydrolysis treatment under non-isothermal regime at a severity of 4.08 (corresponding to a maximum temperature of 210 °C). A liquid to solid ratio of 6 g of liquid/g of solid was employed in order to acquire a concentrated stream of hemicellulosic derived compounds, saving energy and water in the processing [41].

After the hydrothermal treatment, 69.8 g of pretreated Paulownia/100 g of raw Paulownia wood was recovered, which was mainly composed by glucan (54.5 g/100 g of

autohydrolyzed Paulownia wood) and lignin (36.5 g/100 g of autohydrolyzed Paulownia wood), corresponding to recoveries of 95.7 and 99.8% of these components regarding the raw material composition. On the other hand, xylan was almost completely solubilized, remaining only 2.83 g/100 g of autohydrolyzed Paulownia wood.

Concerning the liquid phase, the chemical composition is displayed in Table 1 (noted as stream A). Autohydrolysis liquor yielded 13.0 g of xylooligosaccharides/L and 4.12 g of xylose/L, which means a recovery of 64.2% of xylan regarding the raw Paulownia wood. In this way, the use of high-solid loadings in the pretreatment is of great interest to obtain a more concentrated product reducing operational costs [42]. Nevertheless, undesired compounds for the fermentation as furans (furfural and hydroxymethylfurfural), weak acids (acetic acid), and phenolic compounds are also concentrated, playing the role of inhibitors of the microorganisms responsible of the fermentation [5,17,43]. Consequently, the autohydrolysis liquor was composed by 4.48 g acetic acid/L, 0.28 g hydroxymethyl-furfural/L, and 1.14 g furfural/L. These data were in line with the values reported for autohydrolysis liquor of *Paulownia tomentosa* at a LSR of 8 g/g, with the exception of the acetic acid which is 1.65-fold higher than the value described by Domínguez et al. [44]. Consequently, measurements for the removal of these degradation compounds have to be taken in order to impede their negative effect on the fermentation of hardwood hydrolysates [6].

**Table 1.** Chemical composition in g/L of autohydrolysis liquor (stream A), and hydrolysate after evaluation of acid posthydrolysis (stream B), vacuum evaporation concentration (stream C), overliming (stream D), activated charcoal (stream E) and ion exchange (stream F).

| Components | Stream A | Stream B | Stream C | Stream D | Stream E | Stream F |
|---|---|---|---|---|---|---|
| Glucooligosaccharides | 1.60 ± 0.04 | - | - | - | - | - |
| Xylooligosaccharides | 13.0 ± 0.75 | - | - | - | - | - |
| Acetyl groups | 2.87 ± 0.03 | - | - | - | - | - |
| Glucose | 1.09 ± 0.06 | 3.15 ± 0.16 | 9.86 ± 0.493 | 5.75 ± 0.29 | 5.11 ± 0.26 | 4.34 ± 0.22 |
| Xylose | 4.12 ± 0.12 | 20.14 ± 0.81 | 60.69 ± 2.43 | 61.98 ± 2.48 | 59.95 ± 2.40 | 52.19 ± 2.09 |
| Arabinose | 0.19 ± 0.01 | 0.59 ± 0.02 | 1.62 ± 0.049 | | | - |
| Formic Acid | 0.01 ± 0.00 | 0.02 ± 0.00 | 0.71 ± 0.032 | 0.29 ± 0.01 | 0.18 ± 0.01 | 0.05 ± 0.00 |
| Acetic Acid | 4.48 ± 0.013 | 7.28 ± 0.18 | 5.67 ± 0.142 | 2.95 ± 0.07 | 2.48 ± 0.06 | 0.26 ± 0.07 |
| Levulinic Acid | 0.04 ± 0.00 | 0.12 ± 0.01 | 1.03 ± 0.038 | 0.87 ± 0.03 | 0.33 ± 0.01 | - |
| HMF | 0.28 ± 0.01 | 0.22 ± 0.01 | 0.69 ± 0.009 | 0.11 ± 0.03 | 0.05 ± 0.00 | - |
| Furfural | 1.14 ± 0.05 | 1.16 ± 0.06 | 0.65 ± 0.033 | 0.10 ± 0.01 | 0.05 ± 0.01 | 0.04 ± 0.00 |
| Total phenolic compounds (expressed as gallic acid equivalents) | 3.28 ± 0.00 | 3.05 ± 0.01 | 8.25 ± 0.002 | 0.65 ± 0.00 | 0.32 ± 0.01 | 0.21 ± 0.02 |

*3.2. Evaluation of Dilute Acid Hydrolysis Conditions on the Hemicellulosic Hydrolysate*

Dilute acid hydrolysis with sulfuric acid is a reliable and non-expensive method to obtain xylose-enriched hydrolysate that has a great potential in order to exploit the saccharides from a hemicellulosic hydrolysate. For xylooligosaccharides depolymerization into xylose, range of conditions for dilute acid hydrolysis with $H_2SO_4$ were chosen based on previous works by the authors (such as *Eucalyptus globulus* or corncob). In this case, Garrote and collaborators evaluated the acid posthydrolysis of xylooligosaccharides from *E. globulus* and corn cob liquors after autohydrolysis, studying their hydrolysis by a kinetic modelling [22,45].

Consequently, conditions of 0.5% *w/w* $H_2SO_4$ at 125 °C and 1.5% *w/w* $H_2SO_4$ at 115 °C were chosen for the production of xylose from the hemicellulosic hydrolysate of Paulownia wood. Figure 2 shows the hydrolysis profile of oligomers, monomers, and degradation products at the selected conditions. As the figure displays, both conditions lead to a rapid conversion of the oligosaccharides into monosaccharides. Nevertheless, 1.5% *w/w* $H_2SO_4$ at 115 °C reflected a more appropriate kinetic for faster production of xylose owing to the faster conversion. With this condition, 83% of the xylooligosaccharides were hydrolyzed into xylose at 30 min (see Figure 2c). On the other hand, only 63% of the

xylose was obtained at 105 min from the acid hydrolysis with 0.5% *w/w* $H_2SO_4$ at 125 °C, showing significant differences, $p < 0.02$ (see Figure 2a), and at 160 min, xylose started to degrade into furfural. Using similar conditions of acid hydrolysis 0.5% *w/w* $H_2SO_4$ at 130 °C for 210 min, Moldes and collaborators reached a maximal conversion of the hemicellulosic hydrolysate from autohydrolysis of vine trimming at 190 °C [24]. Regarding degradation compounds, furfural achieved a maximal concentration of 1.38 g/100 g of raw material when employing the conditions of 125 °C, 0.5% $H_2SO_4$ and 160 min. Acetic acid was the second most important compound present in hemicellulosic hydrolysate, and conversely, it was rapidly hydrolyzed at 115 °C with 1.5% $H_2SO_4$ and 30 min.

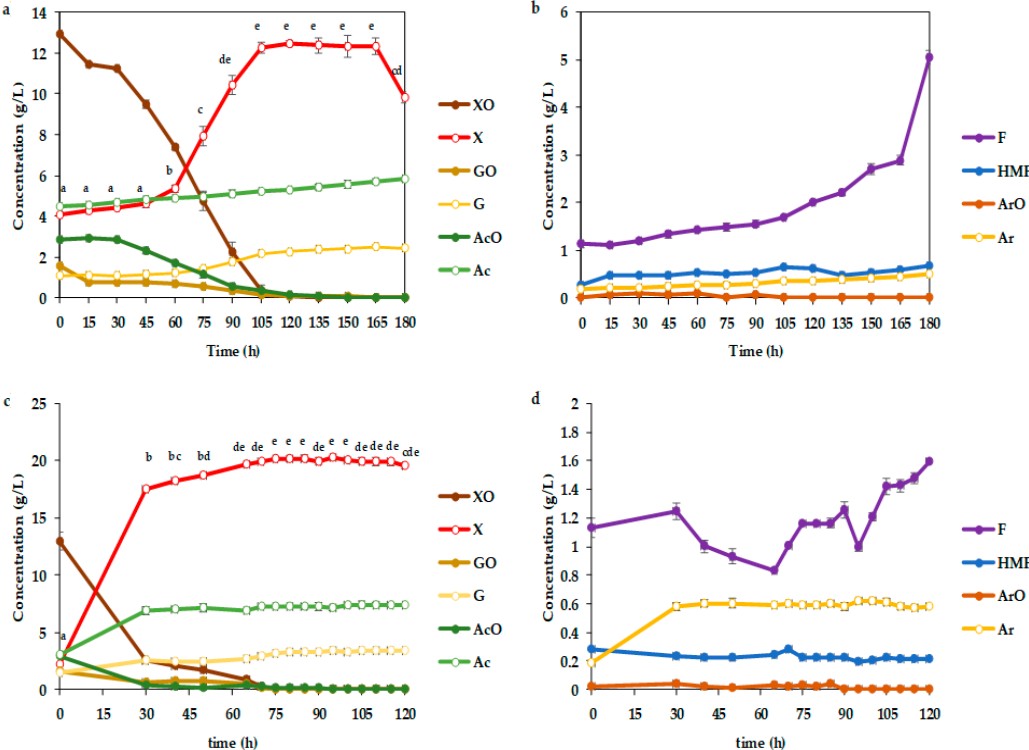

**Figure 2.** Hydrolysis profiles for the acid hydrolysis of hemicellulosic hydrolysates from autohydrolysis of Paulownia wood at 210 °C. (**a**,**b**) represent the time course for the conditions 0.5% *w/w* $H_2SO_4$ at 125 °C and (**c**,**d**) for 1.5% *w/w* $H_2SO_4$ at 115 °C. Figures represent the concentration (g/L) and standard deviation (error bars) of xylooligosaccharides (XO), xylose (X), glucooligosaccharides (GO), glucose (G), acetyl groups linked to oligomers (AcO), acetic acid (Ac), furfural (F), hydroxymethylfurfural (HMF), arabino-oligosaccharides (ArO) and arabinose (Ar). Different letters indicate significant differences ($p < 0.02$) in the xylose values.

In summary, the maximal hydrolysis of xylooligosaccharides into xylose (measured as xylose equivalent/100 g of Paulownia) was obtained with 1.5% $H_2SO_4$ at 115 °C for 75 min, displaying significant differences ($p < 0.02$) regarding the xylose concentration at other reaction times. Therefore, this condition was selected as the most suitable to obtain the hemicellulosic hydrolysate. The complete composition of above mention hydrolysate after the acid hydrolysis is shown in Table 1 (stream B), reaching a concentration of 20 g xylose/L and high concentration of inhibitory compounds such as acetic acid (7.28 g/L).

### 3.3. Concentration and Detoxification of Hemicellulosic Hydrolysate

In order to increase the xylose concentration by three times, vacuum evaporation was carried out, with a consequent removal of volatile compounds. In this sense, the vacuum evaporation could contribute to the detoxification process [46]. In this context, hemicellulosic hydrolysate was detoxified following the successive steps shown in Figure 1. As can be seen in Table 1, xylose was concentrated up to 60.7 g/L (see stream C in Table 1),

with no degradation during the process. On the other hand, acetic acid concentration was reduced to 5.67 g/L, corresponding to a reduction of 74% measured as g of acetic acid/g of hydrolysate in stream B. In addition, furfural also decreased after the vacuum evaporation step to 0.65 g/L, almost half of its value regarding the initial hemicellulosic hydrolysate. Nevertheless, hydroxymethylfurfural was concentrated three times. Similar behavior was reported for the detoxification of eucalyptus and corn stover hemicellulosic hydrolysates by vacuum evaporation [46,47].

Further steps of detoxification were carried out to remove phenolic compounds, furans, and acetic acid. Firstly, the increase of pH to 9–10 in the overliming process (see stream D in Table 1) aided the precipitation of the toxic compounds, and it is proposed as an economic method for hydrolysates detoxification [48]. In fact, hydroxymethylfurfural and furfural were almost removed from the hemicellulosic hydrolysate, remaining in concentrations of around 0.10 g/L. Additionally, significant reduction of acetic acid and formic acid concentrations were observed. On the other hand, regarding sugars, glucose concentration decreased to 5.75 g/L while xylose maintained almost invariable. These results can be compared to those reported by Srilekha Yadav et al., in which similar strategy of detoxification was used for ethanol production from rice straw [32].

Subsequently, activated charcoal process favored the almost total reduction of furans and phenolic compounds, whereas acetic acid stayed in the hydrolysate at a concentration of 2.5 g/L. In this context, concentrations of acetic acid of around 2–5 g/L have been reported as inhibitory for the suitable performance of ethanologenic yeast due to osmotic problems [49]. With the aim to removing completely the acetic acid that may impede the subsequent fermentation, the hydrolysate was submitted to ion exchange process. This procedure allows the recovery of acetate in a separated and purified stream, which could contribute to the economy of the process [25]. As seen in Figure 1, stream F, the inhibitory load (including acetic acid) was highly reduced after this las detoxification step, although 13% of xylose was also removed.

Overall, the resulting concentrated hydrolysates, either detoxified (stream F) or non-detoxified (stream C), were employed for further fermentation assays using *S. stipitis* and recombinant *S. cerevisiae* MEC1133 yeasts (as shown in the following section).

### 3.4. Hydrolysates Fermentation for Ethanol Production

Owing to the amount of compounds (especially acetic acid) in the concentrated hydrolysate (stream C) that may impede the fermentation, different dilutions were assessed as fermentation media to reduce the effect of the inhibitors, namely 100, 80, 60, and 50% of the hydrolysate.

Table 2 displays the main results for the different fermentation experiments performed and Figure 3 shows the fermentation profiles for the non-detoxified (diluted or not) and detoxified hydrolysates using *S. stipitis* CECT 1922 and the recombinant *S. cerevisiae* MEC1133 yeast strain.

As already explained, the dilutions of the non-detoxified hydrolysate (Figure 3a–d) were evaluated in order to reduce the inhibitory effect caused mainly by the high concentration of acetic acid. On the other hand, detoxified hydrolysate was also assessed with both yeast for comparative purposes (Figure 3e).

In light of the data, clear differences can be observed in the ethanol performance by the two yeast, highlighting the incapacity of *S. stipitis* to ferment the non-detoxified concentrated hydrolysate (Figure 3a). Conversely, recombinant industrial strain MEC1133 was able to consume 77% of xylose of non-detoxified hydrolysate at 78 h in the presence of 5.67 g acetic acid/L. In addition, this yeast yielded an ethanol concentration of 10.62 g/L at 48 h, corresponding to 0.37 g/g of ethanol yield.

**Table 2.** Fermentation results for the different options of hemicellulosic hydrolysate media, namely non-detoxified hydrolysate at percentages of 100% (a), 80% (b), 60% (c), and 50% (d), and detoxified hydrolysate at 100% (e). Data show the initial and final concentrations of xylose ($[X]_0$, $[X]_f$), maximum ethanol concentration and yield (($Et)_{MAX}$, $EtYield_{MAX}$) and volumetric productivity at 24 h ($Q_{P24h}$), along with standard deviations. Different letters indicate significant differences ($p < 0.02$) for $(Et)_{MAX}$ and $EtYield_{MAX}$ for each strain.

| Hydrolysate Media Abbreviation | Yeast | $[X]_0$ (g/L) | $[X]_f$ (g/L) | $[Et]_{MAX}$ (g/L) | $EtYield_{MAX}$ ($g_{et}/g_{X0-Xf}$) * | $Q_{P\,24h}$ (g/L·h) |
|---|---|---|---|---|---|---|
| a | | $57.0 \pm 0.10$ | $46.5 \pm 0.60$ | $1.92 \pm 0.14$ [a] | $0.18 \pm 0.01$ [a] | 0.00 |
| b | | $43.8 \pm 0.05$ | $11.7 \pm 0.11$ | $10.2 \pm 0.45$ [b] | $0.32 \pm 0.01$ [b,c] | 0.00 |
| c | *S. stipitis* CECT 1922 | $39.9 \pm 0.73$ | $3.56 \pm 0.12$ | $9.16 \pm 0.27$ [b] | $0.25 \pm 0.00$ [a,b] | 0.00 |
| d | | $27.5 \pm 0.18$ | $3.83 \pm 0.17$ | $8.74 \pm 0.14$ [b] | $0.37 \pm 0.03$ [c] | 0.18 |
| e | | $46.8 \pm 0.79$ | $1.19 \pm 0.02$ | $14.2 \pm 0.40$ [c] | $0.31 \pm 0.01$ [b,c] | 0.53 |
| a | | $55.8 \pm 0.53$ | $12.3 \pm 0.03$ | $14.2 \pm 0.31$ [c] | $0.33 \pm 0.02$ [a] | 0.40 |
| b | | $42.8 \pm 1.67$ | $9.35 \pm 0.61$ | $11.9 \pm 0.21$ [b] | $0.36 \pm 0.03$ [a] | 0.34 |
| c | *S. cerevisiae* MEC 1133 | $37.1 \pm 2.27$ | $4.12 \pm 0.61$ | $10.9 \pm 0.29$ [b] | $0.33 \pm 0.03$ [a] | 0.37 |
| d | | $27.6 \pm 0.97$ | $2.51 \pm 0.00$ | $7.95 \pm 0.05$ [a] | $0.32 \pm 0.04$ [a] | 0.31 |
| e | | $47.6 \pm 1.27$ | $0.18 \pm 0.05$ | $12.5 \pm 0.52$ [b,c] | $0.26 \pm 0.01$ [a] | 0.51 |

* calculated as g of ethanol per g of consumed xylose.

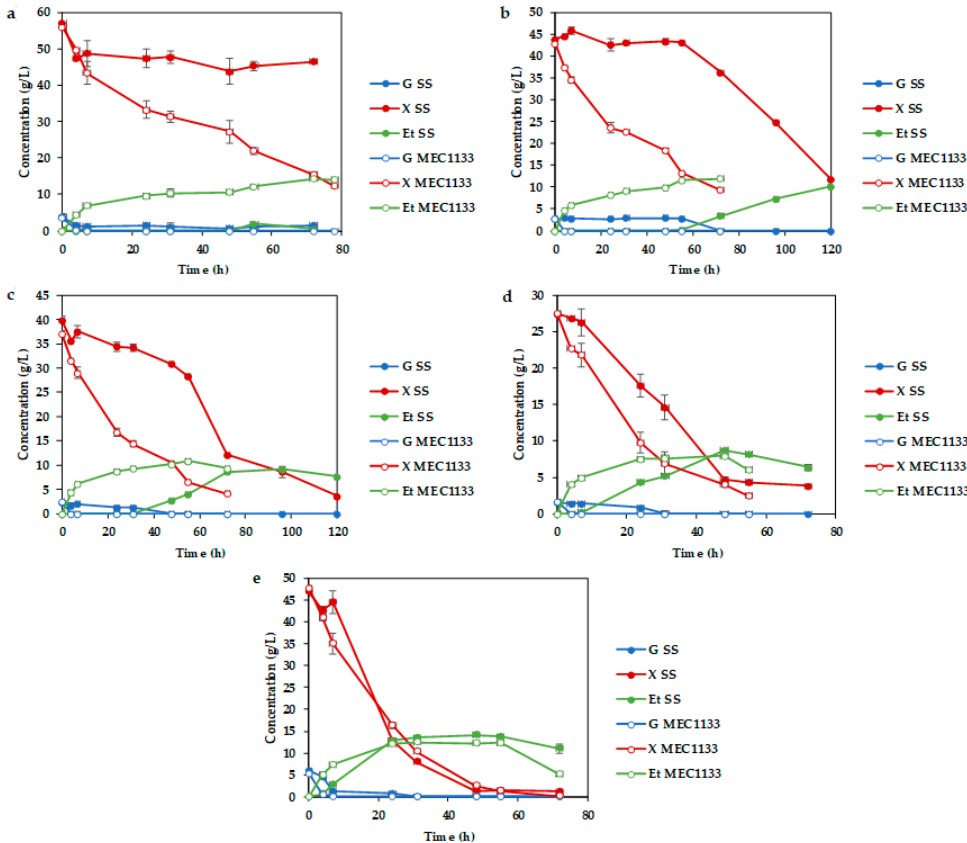

**Figure 3.** Time course of sugar consumption, glucose (G) and xylose (X), and ethanol (Et) production for media of non-detoxified hydrolysate at percentages of (**a**) 100%, (**b**) 80%, (**c**) 60%, and (**d**) 50%, and detoxified hydrolysate at (**e**) 100%, for *S. stipitis* (SS) and recombinant *S. cerevisiae* (MEC1133).

On the other hand, diluted non-detoxified hydrolysate enabled the fermentation of xylose by *S. stipitis*, although there were great dissimilarities. For instance, a 55-h-lag phase was observed when using this yeast with the hydrolysate at a percentage of 80% (Figure 3b). In fact, similar behaviors occurred with the hydrolysates at 60 (Figure 3c) and 50% (Figure 3d) of concentration, with a 31 h a 7 h lag phase, respectively. In this

context, Zhu and collaborators previously reported [50] a direct correlation among the inhibitors concentration (such as acetic acid, hydroxymethylfurfural, furfural and phenolic compounds), and the lag phase in the production of bioethanol by *S. stipitis* is observed. The lag phase is longer for the hydrolysates with higher the inhibitors content.

Contrariwise, recombinant industrial strain MEC1133 did not experience lag phase of any type. Actually, the ethanol course was intense during the first 7 h, leading to concentrations of 4.97–7.38 g/L, which corresponded to $Q_{P7h}$ of 0.71–1.05 g/(L·h). In addition, similar maximum ethanol yields (between 0.26–0.36 g/g) were obtained for the MEC1133 strain regardless the hydrolysate employed for the fermentation, showing no significant differences (for 98% confidence). Conversely, great variations (0.18–0.37 g/g) were found for *S. stipitis* displaying significant differences ($p < 0.02$). Moreover, a maximum ethanol concentration of 14.2 g/L was reached with recombinant industrial strain MEC1133 using the non-detoxified hydrolysate at a percentage of 100%. In this sense, the MEC1133 resulted a suitable yeast for the production of hemicellulosic ethanol without the need of detoxifying the hydrolysate.

In order to get a wider point of view regarding the hemicellulosic ethanol production, Table 3 collects the main results of fermentation on hemicellulosic hydrolysates reported in the literature.

**Table 3.** Comparison of fermentation results using different hemicellulosic hydrolysates.

| Raw Material | Pretreatment | Posthydrolysis and/or Detoxification | Strains | $[X]_0$ (g/L) | $[Et]_{MAX}$ (g/L) | $EtYield_{MAX}$ ($g_{et}/g_{sugar}$) | Refs. |
|---|---|---|---|---|---|---|---|
| Palm press fiber | Acid pret. SL * 30%, 5% $H_2SO_4$, 121 °C, 60 min | - Overliming: $Ca(OH)_2$ and $H_2SO_4$ | *S. stipitis* NRRLY 7124 | 18.6 | 6.13 | 0.33 | [51] |
| *Eucalyptus grandis* | 2% (*v/v*) green liquor ($Na_2S$, NaOH, $Na_2CO_3$) pret. LSR * 3.5 (*w/w*), 155–160 °C for 150 min | - PH *: 4% $H_2SO_4$ (*w/w*), 121 °C, 1 h <br> - Vacuum concentration <br> - Ethyl acetate extraction <br> - Overliming: $Ca(OH)_2$ and $H_2SO_4$ | *S. stipitis* NBRC 10063 | 19.1 | 5.00 | 0.21 ** | [14] |
| Sugarcane bagasse | Hydrothermal pret. SL * 9% (*w/w*), 190 °C, 10 min at 150 rpm | - PH *: 0.5% $H_2SO_4$, 120 °C, 70 min <br> - Vacuum concentration <br> - Overliming: $Ca(OH)_2$ and $H_3PO_4$ <br> - Activated charcoal | *S. stipitis* NRRLY 7124 | 33.5 | 10.6 | 0.32 | [52] |
| Sugarcane bagasse | Hydrothermal pret. SL * 9% (*w/w*), 190 °C, 10 min at 150 rpm | - PH *: 2% $C_4H_4O_4$, 120 °C, 95 min <br> - Vacuum concentration <br> - Overliming: $Ca(OH)_2$ and $H_3PO_4$ <br> - Activated charcoal | *S. stipitis* NRRLY 7124 | 35.1 | 10.6 | 0.30 | [52] |

**Table 3.** *Cont.*

| Raw Material | Pretreatment | Posthydrolysis and/or Detoxification | Strains | $[X]_0$ (g/L) | $[Et]_{MAX}$ (g/L) | EtYield$_{MAX}$ ($g_{et}/g_{sugar}$) | Refs. |
|---|---|---|---|---|---|---|---|
| Sorghum stalks | Acid pret. SL 5% (w/v), 0.2 M $H_2SO_4$, 121 °C, 120 min | - Overliming: $Ca(OH)_2$ and $H_2SO_4$ | *S. stipitis* NCIM 3948 (CBS 6054) | 20.0 | 11.6 | 0.46 ** | [53] |
| Cotton stalks | Alkali pret. SL * 10% (w/v), 3% NaOH; room temperature, 24 h | - Sequential PH *:1% $H_2SO_4$, 110 °C for 30 min and 3% $H_2SO_4$, 130 °C for 60 min <br> - Overliming: $Ca(OH)_2$ and $H_2SO_4$ <br> - Activated charcoal | *S. stipitis* NCIM 3948 (CBS 6054) | 29.4 | 10.1 | 0.45 ** | [54] |
| Exhausted olive pomace | Water extraction at 100 °C 30 min, and acid pret. 2% $H_2SO_4$ 170 °C | - Overliming | *E. coli* SL100 | 23.6 | 13.6 | 0.47 ** | [55] |
| Exhausted olive pomace | Water extraction at 100 °C 30 min, and acid pret. 2% $H_2SO_4$ 170 °C | - Activated charcoal | *E. coli* SL100 | 23.3 | 14.5 | 0.46 ** | [55] |
| Sweet sorghum bagasse | Alkaline treatment and distillation | - | *S. cerevisiae* SFA1$^{OE}$ | - | 17.77 | 0.49 ** | [56] |
| Olive tree pruning | 1% Phosphoric-acid-catalyzed steam explosion (195 °C for 10 min) | - Ion exchange resins | *S. cerevisiae* F12 | 15.9 | 7.5 | 0.32 | [58] |
| Sugarcane bagasse | Supplied by University of São Paulo | - -Undetoxified hydrolysate | Encapsulated GSE16-T18S.1 (T18) *S. cerevisiae* | - | - | 0.38 | [57] |
| *Paulownia elongata x fortunei* | Hydrothermal pret.: LSR * 6 g/g, 210 °C, 150 rpm | - PH *: 1.5% $H_2SO_4$, 115 °C, 75 min <br> - Vacuum concentration <br> - Overliming: BaOH and $H_2SO_4$ <br> - Activated charcoal <br> - Ion exchange resins | *S. stipitis* CECT 1922 | 46.8 | 14.2 | 0.31 | This work |
| *Paulownia elongata x fortunei* | Hydrothermal pret.: LSR * 6 g/g, 210 °C, 150 rpm | - PH *: 1.5% $H_2SO_4$, 115 °C, 75 min <br> - Vacuum concentration <br> - Overliming: BaOH and $H_2SO_4$ | *Recombinant S. cerevisiae* MEC 1133 | 55.8 | 14.2 | 0.33 | This work |

* Abbreviations: SL, solid loading; PH, posthydrolysis; LSR, liquid to solid ratio. ** Calculated regarding the reducing sugars.

Concerning the data from Table 3, initial xylose concentrations between 19.1 and 35.1 were used for fermentation with *S. stipitis* after vacuum concentration of hemicellulosic hydrolysates. In general, hydrolysates from lignocellulosic byproducts, such as palm press fiber [51], sugarcane bagasse [52], sorghum stalks [53], or cotton stalks [54] achieved ethanol yields between 0.30 and 0.46 g ethanol/g sugar. However, fermentation from *E. grandis* hydrolysate enabled an ethanol yield of 0.21 g ethanol/g sugar [14]. In contrast, López-Linares et al. [55] reached ethanol yields of 0.46–0.47 g/g when employing exhausted olive pomace hydrolysate with the bacteria *E. coli* SL100, with a maximum ethanol concentration of 13.6–14.5 g/L. In the present work, an ethanol yield of 0.31 g ethanol/g consumed sugar was obtained when using the *S. stipitis* strain, while the recombinant *S. cerevisiae* MEC1133 produced 0.33 g ethanol/g consumed sugar. Similar results were obtained by metabolic engineered *S. cerevisiae* strains using lignocellulosic hydrolysates, although evolutionary engineering had to be also used [56,57]. Bearing in mind the maximum ethanol concentration, values of 14.2 g/L were obtained in this study, regardless the employed yeast. These data are in accordance or can be positively compared to those listed in Table 3 for yeast fermentation (5.00–11.6 g/L).

As a summary, the use of a detoxified hemicellulosic hydrolysate allowed the production of 14.2 g ethanol/L when employing the yeast *S. stipitis*, with a similar ethanol yield to other reported in the literature (see Table 3). On the other hand, the recombinant industrial strain of *S. cerevisiae* MEC1133 was able to metabolize the hemicellulosic sugar from a non-detoxified concentrated hydrolysate, producing the same amount of ethanol. This work entails an advance towards the production of hemicellulosic ethanol from hardwood hydrolysates and contributes to the study of proper detoxification procedures.

## 4. Conclusions

The process proposed in this work (including biomass pretreatment and liquor processing) enabled to produce ethanol from hemicellulosic fraction of Paulownia biomass showing significant differences in ethanol concentration, yield, and productivity with respect to the yeast used in the fermentation stage. Results verified that natural xylose-consuming yeast strain *S. stipitis* was unable to produce ethanol from non-detoxified hydrolysate, while the engineered xylose-consuming industrial *S. cerevisiae* MEC1133 strain reached 14.2 g ethanol/L. Alternatively, detoxified hydrolysate was appropriate for the manufacture of 14.2 g ethanol/L from *S. stipitis*. Overall, this work shows the viability of using hemicellulose-derived liquors for ethanol production, which could contribute to the integral valorization of polysaccharides present in the fast-growing Paulownia hardwood (including cellulose) to manufacture this biofuel at a large-scale.

**Author Contributions:** Conceptualization, E.D., A.R., G.G., and L.D.; methodology, A.R., G.G., and L.D.; software, E.D., P.G.d.R., and A.R.; validation, A.R., G.G., and L.D.; formal analysis, E.D., P.G.d.R., and A.R.; investigation, E.D. and A.R.; resources, G.G. and L.D.; data curation, E.D., P.G.d.R., and A.R.; writing—original draft preparation, E.D. and A.R.; writing—review and editing, P.G.d.R. and A.R.; visualization, E.D., P.G.d.R.; supervision, A.R., G.G., and L.D.; project administration, A.R., G.G., and L.D.; funding acquisition, G.G. and L.D. All authors have read and agreed to the published version of the manuscript.

**Funding:** This research was funded by MINECO (Spain) in the framework of the projects "Multistage processes for the integral benefit of macroalgal and vegetal biomass" with reference CTM2015-68503-R," and "Cutting-edge strategies for a sustainable biorefinery based on valorization of invasive species" with reference PID2019-110031RB-I00, to Consellería de Cultura, Educación e Ordenación Universitaria (Xunta de Galicia) through the contract ED431C 2017/62-GRC to Competitive Reference Group BV1, program partially funded by European Regional Development Fund (FEDER). This study was also supported by the Portuguese Foundation for Science and Technology (FCT) under the scope of the strategic funding of UIDB/04469/2020 unit.

**Institutional Review Board Statement:** Not applicable.

**Informed Consent Statement:** Not applicable.

**Acknowledgments:** Elena Domínguez would like to express gratitude to the "Programa Iacobus" and to the University of Vigo ("Axudas propias á mobilidade da Universidade de Vigo") for her stay grants. Pablo G. del Río would like to express gratitude to the Ministry of Science, Innovation, and Universities of Spain for his FPU research grant (FPU16/04077).

**Conflicts of Interest:** The authors declare no conflict of interest.

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
