# Peer review of "Hemicellulosic Bioethanol Production from Fast-Growing Paulownia Biomass"

_processes, doi:10.3390/pr9010173_

Round 1

Reviewer 1 Report

In this study, the fast-growing hardwood Paulownia elongata x fortunei was subjected to autohydrolysis processing at selected conditions for the optimal solubilization of hemicellulosic-derived compounds. The obtained results showed that the engineered S. cerevisiae strain produced up to 14.2 g/L of ethanol using the non-detoxified hydrolysate. Nevertheless, the yeast S. stipitis reached similar values of ethanol but only in the detoxified hydrolysate. Consequently, this work enables the valorization of the hemicellulosic fraction of Paulownia wood for the production of second-generation bioethanol.

In my opinion, this article can be publish after minor revision.

Please answer the following comments:

  • Using the full name Saccharomyces cerevisiae Meyen ex E.C. Hansen (Line 43) at least once would be valuable for this article. There is the same situation with other species, e.g. line 47
  • The conclusions lack of the explicit opinion of the authors as to whether this feedstock is a good material for the production of bioethanol and where this plant can be grown or in what large installations its biomass can be used (generally, I mean adding the information about the implementation aspect of your research).
  • It is worth adding a few more references to the discussion.
  • In addition, I reccomend to add a sentence in the conclusions about further research that the authors plan to do connecting to the this raw material, the research methodology etc.

Author Response

In this study, the fast-growing hardwood Paulownia elongata x fortunei was subjected to autohydrolysis processing at selected conditions for the optimal solubilization of hemicellulosic-derived compounds. The obtained results showed that the engineered S. cerevisiae strain produced up to 14.2 g/L of ethanol using the non-detoxified hydrolysate. Nevertheless, the yeast S. stipitis reached similar values of ethanol but only in the detoxified hydrolysate. Consequently, this work enables the valorization of the hemicellulosic fraction of Paulownia wood for the production of second-generation bioethanol.

In my opinion, this article can be publish after minor revision.

Please answer the following comments:

1. Using the full name Saccharomyces cerevisiae Meyen ex E.C. Hansen (Line 43) at least once would be valuable for this article. There is the same situation with other species, e.g. line 47

The authors appreciate the reviewer’s comment, and accordingly the manuscript was modified:

Saccharomyces cerevisiae (Meyen ex E.C. Hansen 1883), is not naturally able to ferment pentoses [8–10]. Among the few yeast strains that can naturally ferment xylose, Scheffersomyces stipitis ((Pignal) Kurtzman & M. Suzuki 2010), …”.

2. The conclusions lack of the explicit opinion of the authors as to whether this feedstock is a good material for the production of bioethanol and where this plant can be grown or in what large installations its biomass can be used (generally, I mean adding the information about the implementation aspect of your research).

We agree with the comment. Although the suitability of the feedstock is exposed in the section 3 (Results and discussion) conclusions were re-written to underline this feature.

“The process proposed in this work (including biomass pretreatment and liquor processing) enabled to produce ethanol from hemicellulosic fraction of Paulownia biomass showing significant differences in ethanol concentration, yield and productivity with respect to the yeast used in the fermentation stage. Results verified that natural xylose-consuming yeast strain S. stipitis was unable to produce ethanol from non-detoxified hydrolysate, while the engineered xylose-consuming industrial S. cerevisiae MEC1133 strain reached 14.2 g ethanol/L. Alternatively, detoxified hydrolysate was appropriate for the manufacture of 14.2 g ethanol/L from S. stipitis. Overall, this work shows the viability of using hemicellulose-derived liquors for ethanol production, which could contribute to the integral valorization of polysaccharides present in the fast-growing Paulownia hardwood (including cellulose) to manufacture this biofuel at a large-scale.”

3. It is worth adding a few more references to the discussion.

The authors appreciate the reviewer’s comment. We tried to use references in order to sustain our hypothesis and results. In this sense, a table with similar works is included in the manuscript with comparative purposes. Nevertheless, more references were included with the aim of completing the manuscript (section 3 and Table 3).

4. In addition, I reccomend to add a sentence in the conclusions about further research that the authors plan to do connecting to the this raw material, the research methodology etc.

As the previous reviewer's comment suggested, conclusions were re-written, including the future perspectives for this work:

“Overall, this work shows the viability of using hemicellulose-derived liquors for ethanol production, which could contribute to the integral valorization of polysaccharides present in the fast-growing Paulownia hardwood (including cellulose) to manufacture this biofuel at a large-scale.”.

Reviewer 2 Report

In my opinion, manuscript ID processes-1074969, although it concerns an interesting and current issue, needs some improvement. My comments below:

  1. The abstract of a good journal paper always has clearly presented aim of research, main results and conclusions, and should be ends outlining the benefits of the study findings and recommendations as a way forward. The manuscript is missing this elements. This should be improve.
  2. Graphical abstract would be very useful for the reader. It would help in understanding the authors' intentions and the scheme of the research work carried out.
  3. The authors should explicitly specify the novelty of their work. What progress against the most recent state-of-the-art similar studies was made in this study? Mention this in the revised manuscript sections, including abstract, introduction, and conclusions. The sentence: “Notwithstanding, the hemicelluloses obtained from the autohydrolysis liquor were not subjected to any further process for the production of bioethanol” is not enough. Please supplement.
  4. The introduction should show the reader more what the authors' research brings to the commonly known knowledge, which inspired them to plan and implement them, and what new they bring to science.
  5. The authors did not formulate any research hypotheses. This should be the starting point for research planning. What did they expect? What were they trying to verify? Needs to be completed.
  6. The purpose of the research misleads the reader. The authors write: "Thus, the aim of this work was the optimization of hemicellulosic hydrolysate ....". Unfortunately, the manuscript did not specify which optimization methods were used. What design methods have been used in the research, eg surfece responce methodology, Palcket-Burmannm method, anova? Have empirical optimization models been developed? And if so, by what methods. This has to be done in the research methodology.
  7. The methodology did not specify with which optimization procedures and statistical methods the significance of differences between the analyzed variables was assessed. It needs to be completed. Without statistical analysis, the discussion of the results is groundless and the conclusions are completely unreliable.
  8. Lack of standard deviation values ​​in the graphs, tables and the content of the manuscript. It is necessary because it gives information about the range of variability of the obtained results.
  9. The conclusions must be verified after proper statistical analysis of the research results.
  10. Author should also pay more attention to the practical implications of this study, outlining the challenges in the current research, future work, and recommendations. Also economic aspect.

Author Response

In my opinion, manuscript ID processes-1074969, although it concerns an interesting and current issue, needs some improvement. My comments below:

  1. The abstract of a good journal paper always has clearly presented aim of research, main results and conclusions, and should be ends outlining the benefits of the study findings and recommendations as a way forward. The manuscript is missing this elements. This should be improve.

Abstract was re-written according to reviewer´s comment.

“In order to exploit a fast-growing Paulownia hardwood as an energy crop, a xylose-enriched hydrolysate was obtained in this work to produce ethanol using the hemicellulosic fraction, that can contribute to the cellulosic valorization of this raw material. For that, Paulownia elongata x fortunei was submitted to autohydrolysis treatment (210 °C or S0 of 4.08) for the xylan solubilization, mainly as xylooligosaccharides. Afterwards, sequential stages of acid hydrolysis, concentration and detoxification were evaluated to obtain fermentable sugars. Thus, detoxified and non-detoxified hydrolysates (diluted or not) were fermented for ethanol production using a natural xylose-consuming yeast, Scheffersomyces stipitis CECT 1922, and an industrial Saccharomyces cerevisiae MEC1133 strain, metabolic engineered strain with the xylose reductase/xylitol dehydrogenase pathway. Results from fermentation assays showed that the engineered S. cerevisiae strain produced up to 14.2 g/L of ethanol (corresponding to 0.33 g/g of ethanol yield) using the non-detoxified hydrolysate. Nevertheless, the yeast S. stipitis reached similar values of ethanol but only in the detoxified hydrolysate. Hence, the fermentation data prove the suitability and robustness of the engineered strain to ferment non-detoxified liquor, and the appropriateness of detoxification of liquor for the use of less robust yeast. In addition, the success of hemicellulose-to-ethanol production obtained in this work shows the Paulownia biomass as suitable renewable source for ethanol production following a suitable fractionation process within a biorefinery approach.”

  1. Graphical abstract would be very useful for the reader. It would help in understanding the authors' intentions and the scheme of the research work carried out.

We agree with the reviewer and a graphical abstract was added to the manuscript to facilitate its understanding.

  1. The authors should explicitly specify the novelty of their work. What progress against the most recent state-of-the-art similar studies was made in this study? Mention this in the revised manuscript sections, including abstract, introduction, and conclusions. The sentence: “Notwithstanding, the hemicelluloses obtained from the autohydrolysis liquor were not subjected to any further process for the production of bioethanol” is not enough. Please supplement.

Abstract and conclusions were re-written according to reviewer´s suggestions. Moreover, introduction was improved to highlight the aim and novelty of the work.

“Abstract: In order to exploit a fast-growing Paulownia hardwood as an energy crop, a xylose-enriched hydrolysate was obtained in this work to increase the ethanol concentration using the hemicellulosic fraction, besides the already widely studied cellulosic fraction. For that, Paulownia elongata x fortunei was submitted to autohydrolysis treatment (210 °C or S0 of 4.08) for the xylan solubilization, mainly as xylooligosaccharides. Afterwards, sequential stages of acid hydrolysis, concentration and detoxification were evaluated to obtain fermentable sugars. Thus, detoxified and non-detoxified hydrolysates (diluted or not) were fermented for ethanol production using a natural xylose-consuming yeast, Scheffersomyces stipitis CECT 1922, and an industrial Saccharomyces cerevisiae MEC1133 strain, metabolic engineered strain with the xylose reductase/xylitol dehydrogenase pathway. Results from fermentation assays showed that the engineered S. cerevisiae strain produced up to 14.2 g/L of ethanol (corresponding to 0.33 g/g of ethanol yield) using the non-detoxified hydrolysate. Nevertheless, the yeast S. stipitis reached similar values of ethanol but only in the detoxified hydrolysate. Hence, the fermentation data prove the suitability and robustness of the engineered strain to ferment non-detoxified liquor, and the appropriateness of detoxification of liquor for the use of less robust yeast. In addition, the success of hemicellulose-to-ethanol production obtained in this work shows the Paulownia biomass as suitable renewable source for ethanol production following a suitable fractionation process within a biorefinery approach.”

“Cellulosic fraction of Paulownia wood was already employed for bioethanol production via a sequential two-stage autohydrolysis [30] and through the combination of autohydrolysis and organosolv [31], or for pulping purposes via soda-antraquinone processing [32]. However, in order to exploit Paulownia wood for energy purposes the employment of the totality of polysaccharides present in this feedstock is essential to improve the feasibility of the process. As far as the authors know, the acid-catalyzed depolymerization of the hemicellulosic fraction into value-added compounds has not been studied.

This study aims to evaluate, for the first time, the acid hydrolysis and fermentation of hemicellulose from Paulownia wood to obtain ethanol. For that, autohydrolysis pretreatment was proposed for xylan solubilization in the liquid phase and several steps of acid xylooligosaccaharides hydrolysis, concentration and detoxification were assessed to improve the ethanol production. Moreover, two yeast (natural xylose consuming and metabolic engineered strains) were used and compared for ethanol production from the hemi-cellulosic non-detoxified and detoxified Paulownia hydrolysates.”

“4. Conclusions

The process proposed in this work (including biomass pretreatment and liquor processing) enabled to produce ethanol from hemicellulosic fraction of Paulownia biomass showing significant differences in ethanol concentration, yield and productivity with respect to the yeast used in the fermentation stage. Results verified that natural xylose-consuming yeast strain S. stipitis was unable to produce ethanol from non-detoxified hydrolysate, while the engineered xylose-consuming industrial S. cerevisiae MEC1133 strain reached 14.2 g ethanol/L. Alternatively, detoxified hydrolysate was appropriate for the manufacture of 14.2 g ethanol/L from S. stipitis. Overall, this work shows the viability of using hemicellulose-derived liquors for ethanol production, which could contribute to the integral valorization of polysaccharides present in the fast-growing Paulownia hardwood (including cellulose) to manufacture this biofuel at a large-scale.”

  1. The introduction should show the reader more what the authors' research brings to the commonly known knowledge, which inspired them to plan and implement them, and what new they bring to science.

The aim of the work was improved in order to clarify the novelty of this study, as seen in revised manuscript. The raw material used in this work is considered a fast-growing biomass with clear benefits to be used as energy crop. Previous studies carried out by the authors showed the adequacy of autohydrolysis treatment to improve enzymatic susceptibility of cellulose and ethanol production from glucose. On the other hand, this work shows the potential of using hemicellulosic fraction to increase the ethanol production using all polysaccharides present in this raw material. Therefore, the present work shows a suitable processing of hemicellulosic hydrolysate to obtain ethanol using a natural xylose-consuming yeast as S. stipitis, and also the advantages of employing an industrial S. cerevisiae metabolic engineered strain able to ferment non-detoxified hydrolysate, avoiding detoxification steps.

  1. The authors did not formulate any research hypotheses. This should be the starting point for research planning. What did they expect? What were they trying to verify? Needs to be completed.

According to reviewer´s comment the abstract, aim and conclusions of the manuscript were re-written in order to clarify the research planning followed and the importance to use the also the hemicellulosic fraction present in this raw material, as indicated previously.

  1. The purpose of the research misleads the reader. The authors write: "Thus, the aim of this work was the optimization of hemicellulosic hydrolysate ....". Unfortunately, the manuscript did not specify which optimization methods were used. What design methods have been used in the research, eg surfece responce methodology, Palcket-Burmannm method, anova? Have empirical optimization models been developed? And if so, by what methods. This has to be done in the research methodology.

We agree with the reviewer. The aim was modified including evaluation of liquors processing instead of optimization. Several conditions (H2SO4 concentration, Temperature and time) of acid hydrolysis of autohydrolysis liquor were evaluated as shown in Figure 2. However, the overliming, ion exchange and activated charcoal stages were carried out based on previous optimized conditions reported in literature.

  1. The methodology did not specify with which optimization procedures and statistical methods the significance of differences between the analyzed variables was assessed. It needs to be completed. Without statistical analysis, the discussion of the results is groundless and the conclusions are completely unreliable.

As replied in the previous question, this study shows an evaluation of acid hydrolysis conditions to obtain xylose from xylooligosaccharides, including sulfuric acid concentration (0.5 and 1-5 %), temperature (125 and 115 ºC) and time of pretreatment (several time points up to 180 min of treatment). Manuscript was reviewed to clarify this point.

  1. Lack of standard deviation values in the graphs, tables and the content of the manuscript. It is necessary because it gives information about the range of variability of the obtained results.

Standard deviation of data shown in Figures and tables were included in the table or figure captions.

  1. The conclusions must be verified after proper statistical analysis of the research results.

Conclusions were accordingly re-written in order to clear up the research results.

  1. Author should also pay more attention to the practical implications of this study, outlining the challenges in the current research, future work, and recommendations. Also economic aspect.

We have revised the manuscript in order to clarify the advantages of this study. As continuation of previous results carried out by our research group, the hemicellulosic fraction of fast-growing Paulownia biomass was evaluated for ethanol production. This approach could increase the potential use of this biomass for biofuel production. Moreover, the use of a robust industrial strain (metabolic engineered with xylose reductase/xylose dehydrogenase pathway) may imply economic benefits since detoxification steps may be avoided.

Round 2

Reviewer 2 Report

Many thanks the Authors for work and improving the manuscript. In my opinion, the manuscript was not properly corrected in research methodology area covering statistical tests which  the authors verified the significance of differences between the analyzed variables (obtained results). Without this information and a properly conducted statistical analysis, the manuscript has very poor scientific value. The values and bars of standard deviations were not introduced either. Alternatively, please include the difference significance comparison tables. This has a great influence on the presentation and interpretation of research results. In my opinion, this must absolutely be corrected.

Author Response

The manuscript was revised according to reviewer´s comment. The deviation standards of results were included in the tables and figures. The section of materials and methods was modified to include statistical analysis in the 2.8 sub-section. Statistical analysis of xylose concentration obtained by acid hydrolysis od xylooligosaccharides was carried out in order to select the most suitable operational conditions (time, temperature, and sulfuric acid concentration) to maximize xylose concentration. Moreover, statistical analysis of ethanol yield and concentration was carried out to show significant differences among results.

Round 3

Reviewer 2 Report

Mnauscript can be publish in present form.